# Characterizations of a Class-I BASIC PENTACYSTEINE Gene Reveal Conserved Roles in the Transcriptional Repression of Genes Involved in Seed Development

Xianjin Ma [1,2,3,†], Yifan Yu [1,2,3,†], Zhikang Hu [1,2,3], Hu Huang [1,2,3], Sijia Li [1,2,3] and Hengfu Yin [1,3,*]

1  State Key Laboratory of Tree Genetics and Breeding, Research Institute of Subtropical Forestry, Chinese Academy of Forestry, Hangzhou 311400, China
2  College of Information Science and Technology, Nanjing Forestry University, Nanjing 210037, China
3  Key Laboratory of Forest Genetics and Breeding, Research Institute of Subtropical Forestry, Chinese Academy of Forestry, Hangzhou 311400, China
*  Correspondence: hfyin@caf.ac.cn; Tel.: +86-571-63105-09320
†  These authors are equally contributed to this work.

**Abstract:** The developmental regulation of flower organs involves the spatio-temporal regulation of floral homeotic genes. *BASIC PENTACYSTEINE* genes are plant-specific transcription factors that is involved in many aspects of plant development through gene transcriptional regulation. Although studies have shown that the *BPC* genes are involved in the developmental regulation of flower organs, little is known about their role in the formation of double-flower due. Here we characterized a Class I BPC gene (*CjBPC1*) from an ornamental flower—*Camellia japonica*. We showed that *CjBPC1* is highly expressed in the central whorls of flowers in both single and doubled varieties. Overexpression of *CjBPC1* in *Arabidopsis thaliana* caused severe defects in siliques and seeds. We found that genes involved in ovule and seed development, including *SEEDSTICK*, *LEAFY COTYLEDON2*, *ABSCISIC ACID INSENSITIVE 3* and *FUSCA3*, were significantly down-regulated in transgenic lines. We showed that the histone 3 lysine 27 methylation levels of these downstream genes were enhanced in the transgenic plants, indicating conserved roles of *CjBPC1* in recruiting the Polycomb Repression Complex for gene suppression.

**Keywords:** BPC transcription factor; flower development; histone methylation; ovule; seed

## 1. Introduction

Floral development requires concerted expression of genes involved in determining organ identity. With the establishment and continuous improvement of the ABC model, the floral homeotic genes, classified into A-, B-, C-, D-, E- categories, have been identified, which forms a complex regulatory network in the control of spatiotemporal gene expression [1,2]. The establishment and maintenance of expression domain of floral homeotic genes are determined by multi-layered regulators [3]; and subtle modification of expression can lead to alterations of floral forms [4].

Transcription factors are often defined as proteins that bind to DNA in a sequence-specific manner and regulate the transcription process [5,6]. In plants, different families of transcription factors are classified based on their specific DNA-binding domains, which play an important role in regulating gene expression in plant growth, development, and stress response [7,8]. The BASIC PENTACYSTEINE (BPC) transcription factors belong to a small plant-specific gene family that contains a highly conserved DNA-binding domain, including five conserved cysteine residues at the C-terminus of proteins [9,10]. BPC genes are found to preferentially bind the "RGARAGRRAA" (GA-rich box or C-box) DNA elements to regulate the expression of downstream genes [9]. In *Arabidopsis thaliana*, *BPC* gene family has seven members; and based on the N-terminus sequences, *BPC* genes can

be divided into three subclasses: Class I contains BPC1/2/3; Class II contains BPC4/6; Class III contains BPC7; BPC5 is likely a pseudogene [10,11]. All *Arabidopsis BPC* genes are ubiquitously expressed in various tissues [12]. It is found that simultaneous mutation of four or five *BPC* genes in *Arabidopsis* cause severe developmental defects in multiple organs, but a weaker phenotype of a single mutation implies functional redundancy among members of the *BPC* family [11,12]. BPC proteins can form homologous and heterologous complexes, indicating that there are complex regulatory relationships between *BPC* genes in the regulation of specific downstream genes [10,13].

Several studies have reported that BPC proteins recruit a high-order protein complex of Polycomb Repressive Complex (PRC) for the suppression of genes through the trimethylation of histone 3 proteins at the lysine 27 position (H3K27me3) [14,15]. Considering that the PRC protein complex does not have the binding ability of specific DNA sites, BPC protein-mediated DNA binding site is crucial in determining the gene expression of floral organ specificity. Indeed, for the refinement of expression domain of C function gene *AGAMAOUS* (*AG*), the GATA motif inside a Polycomb Response Element (PRE)—located in the second intron of *AG*—is recognized by the BPC proteins to recruit the PRC for gene suppression [16,17]. In addition, the *BPC* genes can regulate the expression of HOMEOBOX transcription factors, including *SHOOTMERISTEMLESS* and *BREVIPEDICELLUS*, to maintain the activity of shoot apical meristem. [18]. In the cytokinin signaling pathway, *BPCs* are found to regulate a subset of transcription factors involved in cytokinin actions [19]. In *A. thaliana*, *BPC* genes are initially found to limit the expression of the D function gene (*STK*, *SEEDSTICK*) to regulate ovule development through transcriptional inhibition [9,13]. During the seed development, *BPC* genes are required to determinate the expression of *INNER NO OUTER* (*INO*) [20], *LEAFY COTYLEDON2* (*LEC2*) [21], *ABSCISIC ACID INSENSITIVE 3* (*ABI3*) [22], and *FUSCA3* (*FUS3*) [23].

*Camellia japonica* is an important ornamental flower and possesses different kinds of cultivars with diverse floral variations [24]. The changes of floral pattern and organ shapes in ornamental varieties of *C. japonica* provide a unique resource to interrogate the molecular regulation of floral gene expression [25]. Although the studies have indicated that the *BPC* genes are involved in regulating the expression of floral homeotic genes, it remains unknown how the protein complex mediated by BPCs is regulated to determine the organ-specific expression of floral homeotic genes, particularly in the regulation of double flower formation. In this study, we characterized a Class-I BPC homolog, *CjBPC1*, from *C. japonica*. We showed that the expression of *CjBPC1* is highly expressed in the inner flower tissues in both single and doubled varieties. We demonstrated that *CjBPC1* possessed conserved functions of regulating genes involved in ovule and seed development through the H3K27me3 modifications of downstream genes. Our results indicate *BPC* genes are potential regulators of double flower formation in *C. japonica*.

## 2. Materials and Methods

### 2.1. Plant Materials and Growth Conditions

The *Camellia japonica* L. materials used in this experiment were planted in the Research Institute of Subtropical Forestry, China Academy of Forestry (119°57′ N, 30°04′ E, Fuyang, China). The flower bud materials of *C. japonica* were collected and classified and immediately frozen in liquid nitrogen and stored in a refrigerator at −80 °C. The *Arabidopsis* (Colombia ecotype) was used and maintained in a growth chamber (long-day conditions, 16 h light/8 h dark, set at 22 °C).

### 2.2. Identification and Phylogenetic Analysis of BPC Genes

Seven BPC protein sequences from *A. thaliana* were downloaded from TAIR (https://www.Arabidopsis.org/ (accessed on 10 May 2020)). The software BioEdit software was used to identify BPC sequences using local *C. japonica* datasets through the BLASTP program (E-value cutoff 10–15). The conserved domains of the *BPC* genes were analyzed on the SMART [26] with default settings. Protein sequences from *C. japonica* and *A. thaliana*

are used to construct a phylogenetic tree by the MEGA 7.0 software with parameters described before [27]. The online tool MEME (meme-suite.org) was used to identify the conserved motifs of BPC proteins [28]. The parameters were as follows: the distribution of arbitrary number of repeat sites, the found six motifs, and the minimum and maximum motif widths were six and 50, respectively. The subcellular localization of CjBPC proteins was predicted by CELLO v.2.5 server (http://cello.life.nctu.edu.tw/cello2go/ (accessed on 10 May 2020)) [29,30].

### 2.3. RNA Extraction and qRT-PCR Analysis

Total RNA was extracted using a plant total RNA extraction kit (RNAprep Pure polysaccharide polyphenol, TIANGEN, Beijing, China), and then the first strand of cDNA was synthesized using PrimeScript™ II 1st Strand cDNA Synthesis Kit (Takara, Dalian, China). Real-time quantitative PCR (qRT-PCR) primers were designed by Primer Express 3.0.1 to amplify short fragment DNA (50–100 bp). SYBR® Premix Ex TaqTM II (Takara, Dalian, China) reagents and an ABI 7300 REAL-TIME PCR instrument were used for PCR analysis. Three replicates per test were used for obtaining the gene expression results, and data analysis was based on the $2^{-\Delta\Delta CT}$ method [31].

### 2.4. Vector Construction and Arabidopsis Transformation

The full-length sequences of *CjBPC1* gene were cloned using specific primers (Supplementary Table S1), and cloned into pEXT06/g vector (Baige, Suzhou, China) according to the user's manual. The recombinant plasmid was validated and transformed into *Agrobacterium tumefaciens* C58 (pGV3101) strain by heat shock method, and then used for the transformation of *A. thaliana* by the floral-dip method [32].

### 2.5. Phenotypic Analysis of Transgenic Lines

The transformed *Arabidopsis* T1 seeds were germinated on 1/2 Murashige and Skoog medium containing hygromycin (10 mg/L) for selection. The genomic DNA of transgenic plants was extracted from the young leaves (Two-week old), and PCR validation was performed using hygromycin and construct-specific primers to determine positive transgenic lines. In addition, qRT-PCR was performed to reveal the ectopic expression of *CjBPC1* in transgenic plants. The length and number of seeds of deformed and shortened pods of transgenic lines were measured.

### 2.6. Chromatin Immunoprecipitation PCR (ChIP-PCR)

The ChIP-PCR experiment was performed according to previous studies [33]. The H3K27me3 antibody was obtained from Sigma-Aldrich (Catalogue Number: 07-449). For chromatin isolation and immunoprecipitation, around 1.5 g inflorescence of transgenic *A. thaliana* were taken to extract DNA for PCR analysis. Two independent chromatin isolations were performed, and three PCR amplification replicates were used for quantitative analysis according to $2^{-\Delta\Delta CT}$ method [31].

### 2.7. Statistical Analysis

The one-way analysis of variance (ANOVA) test is used to perform significance test for groups of samples by SPSS software [34]. Different letters indicate the significant differences ($p$-value < 0.01). The unpaired two-tailed Student's tests are used for to determine the statistical significance. * $p < 0.05$; ** $p < 0.01$.

## 3. Results

### 3.1. Identification and Characterization of BPC Genes in C. japonica

To identify BPC family genes, we performed a homology search in the transcriptome sequence database of *C. japonica* [25,35] using sequences of *Arabidopsis* BPC proteins. Six candidates were identified based on the similarity of their coding sequences. Then we cloned the full-length coding sequences of the candidates using gene-specific primers

(Supplementary Table S1) and validated the transcript sequences. The length of deduced amino acids (AAs) from the transcripts ranged from 235 to 332 (Supplementary Table S1). The predicted isoelectric points of all proteins were relatively high ($p > 9$; Supplementary Table S2), largely due to the abundance of alkaline AAs.

To study the sequence characteristics, we performed sequence-alignment analysis by using the *BPC* genes from *C. japonica* and *A. thaliana*. We found all of the six candidates from *C. japonica* contain the highly conserved motif in which the conserved arrangement of the five cysteine residues was revealed (Figure 1A). To gain a deep understanding of the evolution of the *BPC* genes in *C. japonica*, we constructed a phylogenetic tree. We showed that BPC proteins from *C. japonica* were categorized into three groups: two members in the Class I clade, two members in the Class II clade and one member in the Class III clade (Figure 1B). Through sequence alignment and motif analyses, we found that, the subclades of BPCs possessed several conserved motifs in the N-terminal regions (Figure 1A,C). The classification of sub-clades of *BPC* genes is consistent with previous studies [11], suggesting functional conservations of *BPC* genes in different plant lineages.

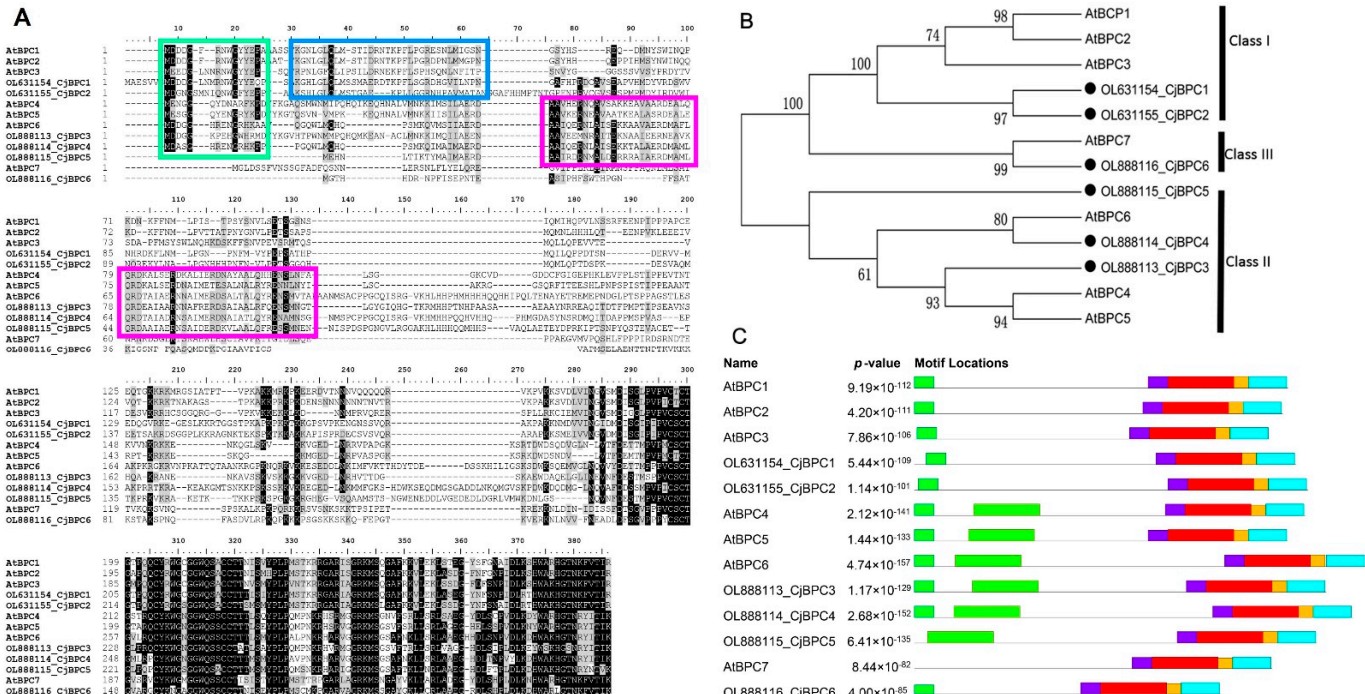

**Figure 1.** Sequence alignment and phylogenetic analysis of BPC family genes from *C. japonica*. (**A**), protein sequence alignment of BPCs from *C. japonica* and *A. thaliana*. Green rectangle indicates the conserved N-terminal sequences in Class I and Class II *BPC* genes; blue rectangle indicates a conserved domain in the Class I *BPC* genes; magenta rectangles indicate the conserved domains in the Class II *BPC* genes. (**B**), a phylogenetic tree containing the *BPC* genes from *C. japonica* and *A. thaliana*. Three classes are indicated by black bars. (**C**), the distribution of motifs that are predicted by SMART analysis. Details of domains are presented in Supplementary Figure S1. Different color bars indicate the consensus sequences of motifs.

### 3.2. Expression Analysis of CjBPC1 in Floral Tissues of Single and Double Flowers of C. japonica

To reveal potential roles of *CjBPC* genes in floral development, we analyzed the expression profiles of *CjBPCs* in different tissue types. We found that the members of *CjBPCs* displayed different overall expression levels: *CjBPC1* and *CjBPC5* were abundantly detected in all tissue types; *CjBPC2*, *CjBPC3*, and *CjBPC6* were relatively lowly expressed; *CjBPC4* was barely detected (Supplementary Figure S2) [35]. *CjBPC1* was highly expressed in floral bud, indicating a potential function of regulating floral development. We further evaluated the *CjBPC1* expression in floral organs of single and doubled flowers. The four

floral tissues types (sepal, petal, stamen and carpel) from wild *C. japonica*, together with floral tissues from anemone double and formal double varieties, were studied (Figure 2A). We found that the expression of *CjBPC1* was detectable in all examined tissues (Figure 2B), and *CjBPC1* displayed higher expression levels in carpels in wild *C. japonica*. In doubled cultivars, *CjBPC1* was markedly up-regulated in the central tissues of doubled flowers (carpel in anemone double and inner petals of formal double) (Figure 2B).

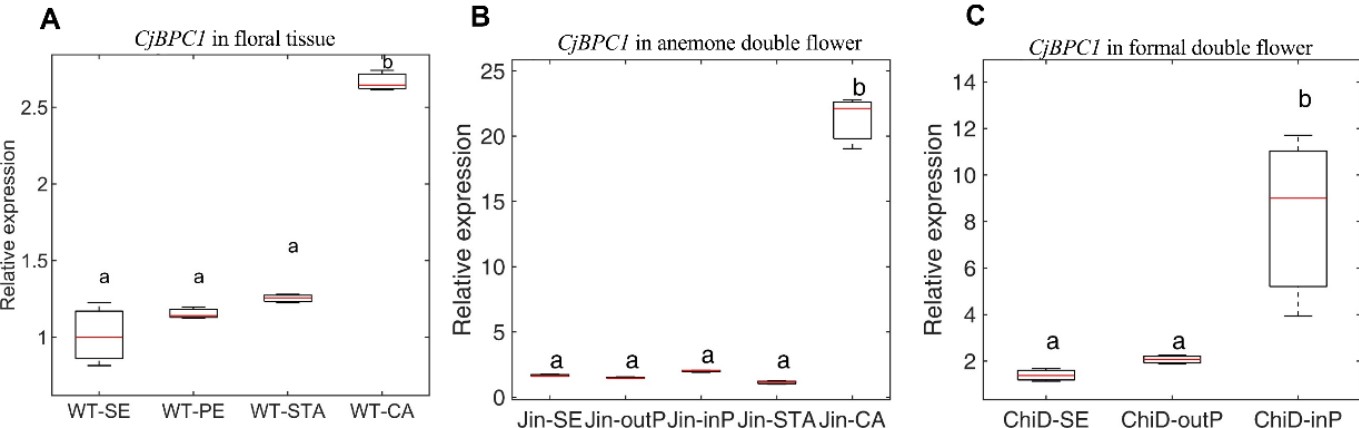

**Figure 2.** Expression analysis of *CjBPC1* in wild and doubled *C. japonica* plants. (**A**), the expression of *CjBPC1* in four floral organs in wild single *C. japonica*. WT-SE, sepal; WT-PE, petal; WT-STA, stamen; WT-CA, carpel. (**B**), the expression of *CjBPC1* in anemone double variety 'Jinpanlizhi'. Jin-SE, sepal; Jin-outP, outer whorl of petal; JIN-inP, inner whorl of petal; Jin-STA, stamen; Jin-CA, carpel. (**C**), the expression of *CjBPC1* in formal double variety 'Chidan'. ChiD-SE, sepal; ChiD-outP, outer whorl of petal; ChiD-inP, inner whorl of petal. Three biological replicates are used for each tissue and two technical replicates of real-time qPCR analysis are performed. The GAPDH gene was used as the internal reference for gene expression analysis (Supplementary Table S1). For each boxplot, the red bar indicates the mean value. Box limits represent the upper and lower quartiles. Whiskers represent minimum to the lower quartiles and maximum to the upper quartiles. One-way ANOVA and multiple comparisons are performed for significance tests (*p*-value < 0.01), and different letters indicate significant changes.

### 3.3. Ectopic Expression of CjBPC1 in Arabidopsis Causes Ovule Abortion

To understand the function of *CjBPC1*, we generated the transgenic *Arabidopsis* lines through ectopic expression of *CjBPC1*. The transgenic lines were verified by PCR amplification and gene expression analysis (Supplementary Figure S3), and we selected five lines with ectopic expression of *CjBPC1* for the phenotypical evaluations (Supplementary Figure S3B,C). Moreover, we showed that the CjBPC1 predominantly localized in the nuclei (Figure 3), suggesting its functions in the regulation of gene expression.

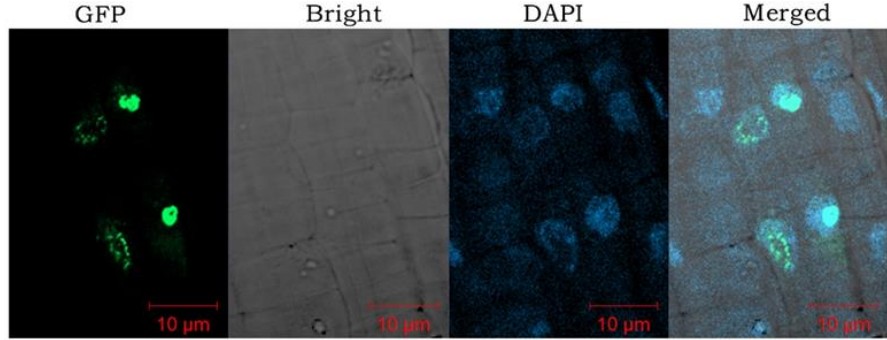

**Figure 3.** Subcellular localization of CjBPC1 in transgenic *A. thaliana* lines. From left to right: the light field, GFP signal, (DAPI 4′,6-diamidino-2-phenylindole) staining of nuclei, merged image. The root tips of transgenic lines are used for confocal microscopy analysis. Bars = 10 μm.

We analyzed the phenotypical alterations using transgenic lines with the high-level expression of *CjBPC1* (line 3 and line 6) (Supplementary Figure S3C). We found no visible morphological changes during the vegetative growth; during the reproductive development, the siliques in transgenic lines had conspicuous defects: the shortened and deformed silique were frequently produced in the inflorescences of transgenic plants (Figure 4A–C). To compare the phenotypical effects, we classified the abnormal siliques into "strong" and "weak" categories by the length of the siliques: strong, less than 6 mm; weak, between 6–10 mm (Figure 4D,E). We found that there were varying degrees of aborted ovules throughout the development of ovules in the five assessed transgenic lines (Figure 4).

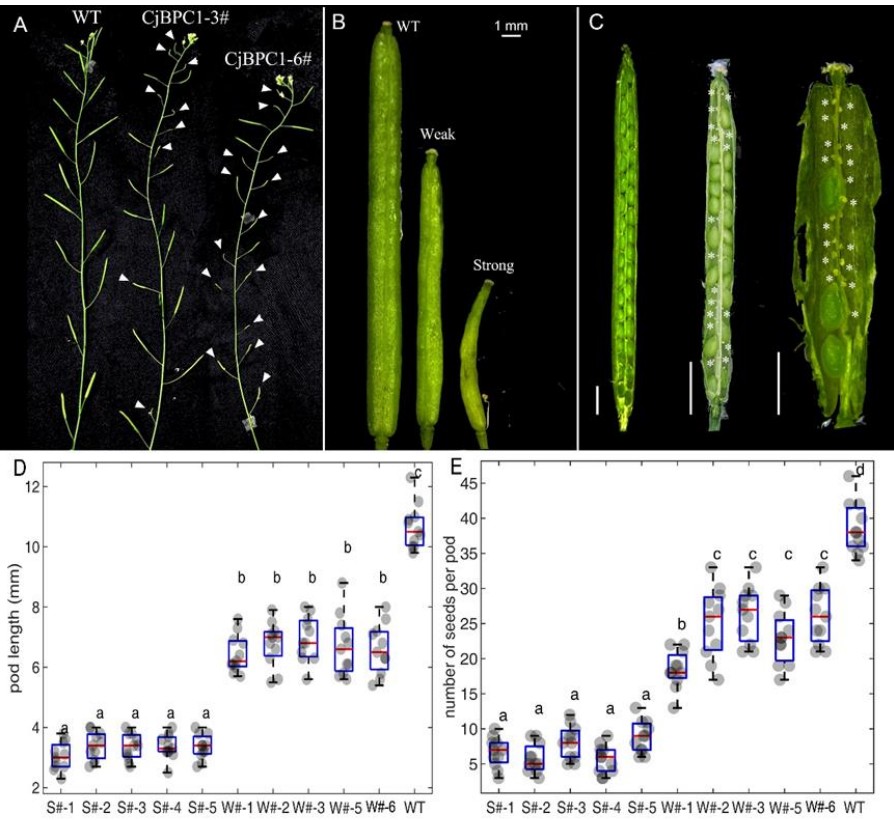

**Figure 4.** Overexpression of *CjBPC1* causes seed abortion in *A. thaliana*. (**A**), the comparison of inflorescences in WT and *CjBPC1* transgenic lines (line CjBPC1-3 and line CjBPC1-6). Arrowheads indicate the shortened or deformed siliques. (**B**), close-up images of siliques, including strong and week phenotypical changes comparing to the WT. Bar = 1 mm. (**C**), the dissected siliques of transgenic lines. From left to right: mature siliques with normal, weakly deformed and strongly deformed phenotypes. The stars indicate the aborted seeds. Bars = 1 mm. (**D**), Measurements of the length of mature pods in different transgenic lines (Line 1, 2, 3, 5, 6) and WT. (**E**), Measurements of the number of seeds per pod. (**D**,**E**), The pods from transgenic lines are categorized into strong (S#) and weak (W#). The one-way ANOVA test is used to perform significance test. Each group contains 11 measurements. Different letters on top of boxes indicate the significant differences ($p$-value < 0.01). For each boxplot, the red bar indicates the mean value. Box limits represent the upper and lower quartiles. Whiskers represent minimum to the lower quartiles and maximum to the upper quartiles. The grey dots indicate the raw values of each measurement.

### 3.4. Overexpression of CjBPC1 Leads to Down-Regulation of Genes Involved in the Seed Development

The *BPC* genes have been found to be involved in several aspects of ovule and seed development through regulating the expression of downstream genes [23]. We investigated the expression of genes that are involved in the seed development using the transgenic lines of *CjBPC1*. The significant down-regulation of *STK*, *LEC2*, *ABI3* and *FUS3* was revealed

in both strong and weak transgenic lines (Figure 5). Consistently, strong lines displayed severer reduction of gene expression than that of weak lines (Figure 5). These results indicate that these genes are probably downstream genes of *CjBPC1* that are responsible for the defects of siliques; and *CjBPC1* might share conserved functions of regulating the expression of genes involved in seed development.

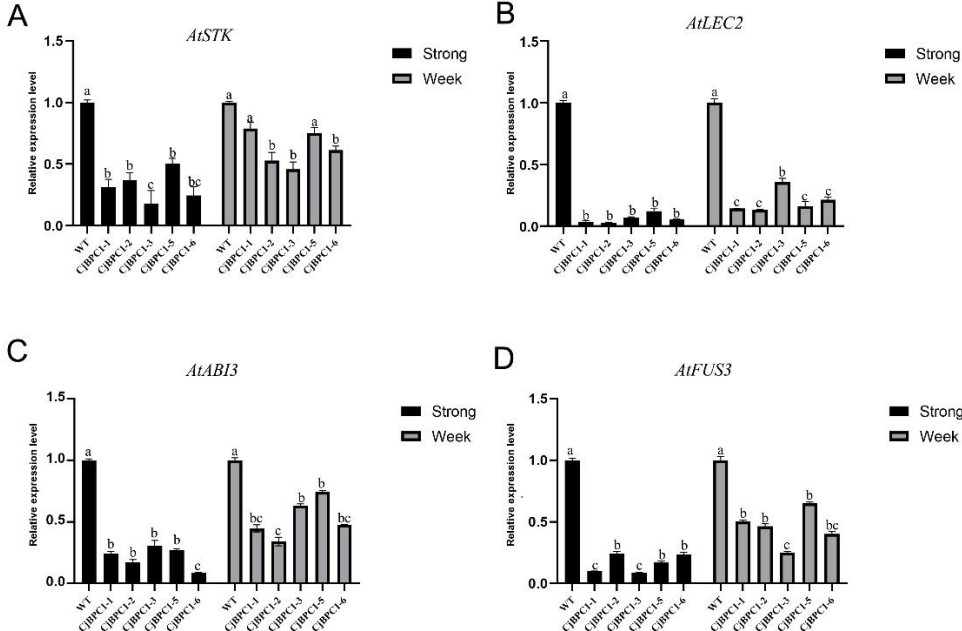

**Figure 5.** Expression analysis of genes involved in seed development from *A. thaliana*. The expression of *STK* (**A**), *LEC2* (**B**), *ABI3* (**C**), *FUS3* (**D**) in different transgenic lines. **A-D**: strong and weak siliques are used for gene expression separately. Three biological replicates of each sample are used, and two technical replicates of real-time PCR experiments are performed to obtain the gene expression levels. *ACTIN* gene was used as the internal reference (Supplementary Table S1). Error bars indicate standard deviations of all the replicates in each genetic background. The one-way ANOVA test is used to perform significance test. Different letters on top of columns indicate the significant differences (*p*-value < 0.01).

### 3.5. Transgenic Lines of CjBPC1 Displays Enhanced H3K27me3 Levels of Downstream Genes

The mechanism of BPC mediated gene repression is involved in the recruitment of PRC complex that leads to the enhanced levels of H3K27me3 [13]. To further investigate if overexpression of *CjBPC1* caused direct changes of histone status of the downstream genes, we investigated the H3K27me3 levels in transgenic *Arabidopsis* lines. We performed the Chromatin Immunoprecipitation PCR (ChIP-PCR) analysis of H3K27me3 at the regions of the downstream genes [21,23,36,37] (Figure 6). We showed that the increased levels of H3K27me3 were identified in all the tested downstream genes at different levels (Figure 6), in which *STK* and *ABI3* had over three folds of methylation levels. These results indicate that *CjBPC1* might be directly involved in the regulation of H3K27me3 levels through recruiting the PRC complex in *A. thaliana*.

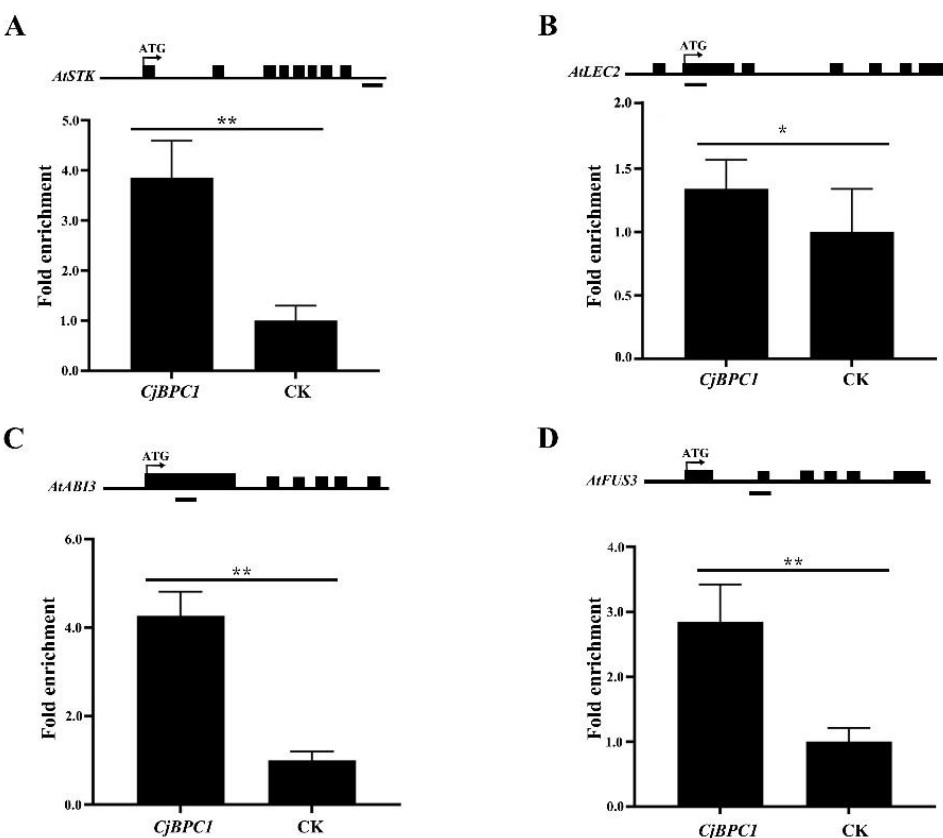

**Figure 6.** Analysis of H3K27me3 levels in WT and transgenic lines of *CjBPC1*. The H3K27me3 level of *STK* (**A**), *LEC2* (**B**), *ABI3* (**C**), *FUS3* (**D**) loci were measured by ChIP-PCR using the fragment indicated on top of each panel (indicated by the black bars under the gene models). The immunoprecipitation experiment was performed twice and three replicates were performed to obtain quantification values. Error bars indicate standard deviations of all the replicates. The student's t tests are used for the comparisons between control group (CK) and transgenic lines. * $p < 0.05$; ** $p < 0.01$.

## 4. Discussion

The *BPC* gene family is unique among plants and usually has a small number of members. By recruiting PRC complexes for their role in gene regulation, *BPCs* may affect far more genes in regulation of many aspects of plant development than has been revealed so far [38]. A recent study based on the ChIP sequencing analysis of *BPC6* uncovered thousands of potential direct binding sites in *Arabidopsis* genome [19]. It is not known, however, whether and how different classes of *BPC* genes acquire sequence-preferences in regulating their downstream genes.

Although, in *A. thaliana*, all members of *BPCs* exhibit ubiquitous expression in different tissue types, the transcriptional regulation of *BPCs* remains an important aspect of functional divergence [11]. We found that, in *C. japonica*, the expression patterns of different *BPCs* differed greatly in terms of the expression abundance, based on transcriptomics analysis using relatively mature organs (Supplementary Figure S1). Particularly, *CjBPC1* (a Class I member) and *CjBPC5* (a Class II member) were highly expressed in all tissues, indicating the major roles of maintaining the gene expression profiles (Figure 1; Supplementary Figure S1). Thus, different members of *BPCs* might play diverse functions at different stages of plant development or in certain specified tissues. The expression level of *CjBPC1* was detected in all floral organs of wild *Camellia* flower and markedly accumulated in the carpel tissues (Figure 2A), and this result was in good agreement with the phenotypic changes observed in transgenic *Arabidopsis* (Figure 3), which indicated that *CjBPC1* is mainly involved in ovule development. However, the expression of *CjBPC1* did not appear to be entirely tissue-specific: in the formal double variety "Chidan", which is

completely devoid of ovule development, *CjBPC1* remained highly expressed in the central regions of floral buds, where ovules in wild *Camellia* differentiate (Figure 2C). Therefore, *CjBPC1* is potentially involved not only in specifying ovule tissues but also in maintaining the floral meristem activity.

Overexpression of different *BPC* members can reveal functional specificity in the regulation of gene expression. For example, ectopic expression of *BPC2* caused down-regulation of *LATE EMBRYOGENESIS ABUNDANT* (*LEA*) genes, which led to enhanced susceptibility of osmotic stress in *A. thaliana* [39]. In rice, overexpression of a Class-II BPC gene revealed divergent functions in regulation flowering time [40]. We showed, in the *Arabidopsis* system, the regulation of ovule/seed development was affected by the heterogeneous expression of *CjBPC1* (Figure 3). This result indicated *CjBPC1* might have specific functions in ovule or seed development. In *Arabidopsis*, the Class-I *BPC* genes were also found to regulate *HOMEOBOX* genes for the meristem maintenance [18]. In *CjBPC1* transgenic lines, because no visible phenotypes at the vegetative stage (e.g., leaf development and flowering time) were observed, it is likely that expression of *HOMEOBOX* genes is not altered. Gene expression analysis of potential downstream genes supported that *CjBPC1* is mainly involved in regulating expression of ovule- and seed-related genes (Figures 5 and 7). We also found that the H3K27me3 levels of downstream genes are enhanced in transgenic lines (Figure 6). These results are consistent with molecular functions of *BPC* genes [13,41]. We postulated that *CjBPC1* share conserved functions of regulating ovule and seed development through regulating the subset of downstream genes in *C. japonica*. We thereby proposed a model of the functions in the transgenic lines of *Arabidopsis* which involved the recruitment of PRC and suppression of genes involved in seed development (Figure 7).

**A Model of BPC-Mediated Suppression of Genes**

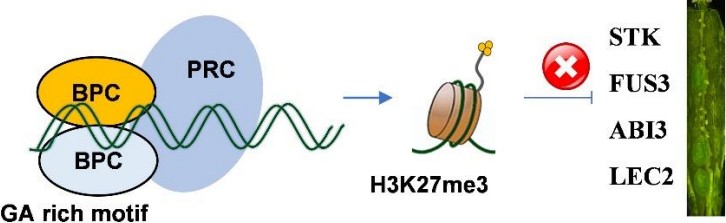

**Figure 7.** A molecular model of *BPC* genes and their roles in the regulation of seed development. The BPC proteins can form hetero- or homo- dimers during the binding of the GA-rich motifs [6,7]; this leads to the tri-methylation of histone 3 lysine 27 and further suppress the expression of genes including *STK*, *FUS3*, *ABI3* and *LEC2*.

The spatiotemporal expression of floral homeotic genes is critical for floral development and is controlled by multi-layered regulators [42,43]. For the regulation of floral homeotic genes, it was found that the *BPC* genes are necessary to limit the expression domain of *AG* and *STK* [9,13,16]. The functional action of *BPCs* mainly involves two aspects: First, the binding to the GA-rich box to regulate downstream genes; Second, interact with proteins to recruit the PRC complexes. Based on a molecular model of PRE mediated gene suppression [17,44], BPCs and GA-rich motif are components of the regulatory mechanism [44]. It is attempting to hypothesize that the BPC-mediated gene suppression is an upstream mechanism for ensuring the correct expression of floral homeotic genes. Double flowers in *C. japonica* exhibits extensive changes in floral organ identity, number of flower organs and arrangement of flower organs; and the alterations in the expression of floral homeotic genes (e.g., B- and C- class genes) have been revealed to be associated with the formation of double flowers [24,45]. Further studies of BPCs and GA-rich motifs could lead to the discovery of potential regulatory mechanisms underlying the modifications of gene expression in double flowers of *C. japonica*.

## 5. Conclusions

In conclusion, this work revealed that *CjBPC1*—the Class-I homolog of BPC family—has conserved functions of gene suppression in regulating flower and seed development; and BPC members from *C. japonica* are potential regulators for establishing and maintaining the expression pattern of floral homeotic genes during double flower formation.

**Supplementary Materials:** The following supporting information can be downloaded at: https://www.mdpi.com/article/10.3390/cimb44090278/s1.

**Author Contributions:** H.Y. conceived the research objectives. X.M., Y.Y. and Z.H. performed gene expression and functional analyses of BPC genes. H.H. and S.L. participated in the sample collection. X.M. and H.Y. drafted the manuscript and all authors contributed and approved the paper. All authors have read and agreed to the published version of the manuscript.

**Funding:** This research is supported by Nonprofit Research Projects (CAFYBB2021QD001-1) of Chinese Academy of Forestry and National Science Foundation of China (31870578).

**Institutional Review Board Statement:** Not applicable.

**Informed Consent Statement:** Not applicable.

**Data Availability Statement:** The data presented in this study are available from the web link of the manuscript.

**Conflicts of Interest:** The authors declare no conflict of interest.

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
