# Peer review of "Characterizations of a Class-I BASIC PENTACYSTEINE Gene Reveal Conserved Roles in the Transcriptional Repression of Genes Involved in Seed Development"

_cimb, doi:10.3390/cimb44090278_

Round 1

Reviewer 1 Report

The titles of sections/subsections should be formatted uniformly.

The information presented in the  descriptions figures should be revised. Some information is repeated. 

The latin names of plants should be checked to be italised.

The list of references should be revised according the recommendations for authors.

Author Response

Reviewer 1:

Comments and Suggestions for Authors

The titles of sections/subsections should be formatted uniformly.

#Thank you for the comment. We have re-formatted the titles of sections in this revised manuscript.

The information presented in the descriptions figures should be revised. Some information is repeated.

#Sorry for the overlook. We have checked the information of figure legends and descriptions and removed repeated contents during this revision. Please find the revised figure legend in this version of manuscript. Thank you.

The latin names of plants should be checked to be italised.

#Sorry for the overlook. We have checked the latin names throughout the manuscript and revised accordingly.

The list of references should be revised according the recommendations for authors.

#Thank you for the comment. We have revised the references information to conform with the format requirements in this revision.

Reviewer 2 Report

Find below my suggestions:

Abstract is very typical. no scientific soundness. Re-write in a scientific manner.

In introduction, add some information about transcription factors (start of second paragraph).

Significance of the study is missing. What are the research gaps?

Materials and Methods: which statistical method used by authors to analyze the data?

Why authors have only given the motifs? Gene structure also include conserved domains and introns/exons. Add intron/exon structure along with the motifs figure.

GFP signal is not very clear. I would suggest authors to use a nuclear marker as a reference.

Moreover, check the location of the current gene from online server (http://cello.life.nctu.edu.tw/cello2go/) followed by functional verification.

Reviewer 3 Report

In the manuscript entitled, Characterizations of a Class-I BASIC PENTACYSTEINE gene reveal conserved roles in the transcriptional repression of genes involved in seed development, authors have functionally characterised Class I BPC gene (CjBPC1) from an ornamental flower—Camellia japonica which is related to regulation of double flower. The authors showed that CjBPC1 is highly expressed in the central whorls of flowers in both single and doubled varieties. Overexpression of CjBPC1 in Arabidopsis thaliana caused sever defects in siliques and seeds.

The manuscript is suitable for the journal and falls under the aim and scope of the journal. It looks reasonable. The article should be accepted after minor revisions.

1)      The gene name mentioned in the figure should be presented in italics.

2)      The motif details should be provided in a supplementary file 

Author Response

Reviewer 3:

Comments and Suggestions for Authors

In the manuscript entitled, Characterizations of a Class-I BASIC PENTACYSTEINE gene reveal conserved roles in the transcriptional repression of genes involved in seed development, authors have functionally characterised Class I BPC gene (CjBPC1) from an ornamental flower—Camellia japonica which is related to regulation of double flower. The authors showed that CjBPC1 is highly expressed in the central whorls of flowers in both single and doubled varieties. Overexpression of CjBPC1 in Arabidopsis thaliana caused sever defects in siliques and seeds.

The manuscript is suitable for the journal and falls under the aim and scope of the journal. It looks reasonable. The article should be accepted after minor revisions.

#Thank you for the comments.

1)      The gene name mentioned in the figure should be presented in italics.

#We have checked the gene name throughout the manuscript and corrected them according to your comment.

2)      The motif details should be provided in a supplementary file 

#The comment is taken. In this revised manuscript, we have added a supplementary figure (the new Supple. Fig. S1) to include the detailed information of motifs. Please find the information in this revised manuscript.